# Childhood Socioeconomic Position, Adult Educational Attainment and Health Behaviors: The Role of Psychological Capital and Health Literacy

**DOI:** 10.3390/ijerph18179399

**Published:** 2021-09-06

**Authors:** Karlijn Massar, Natalie Kopplin, Karen Schelleman-Offermans

**Affiliations:** 1Department of Work & Social Psychology, Faculty of Psychology & Neuroscience, Maastricht University, P.O. Box 616, 6200 MD Maastricht, The Netherlands; karen.offermans@maastrichtuniversity.nl; 2CP Consultingpartner AG, Venloer Straße 53, 50672 Cologne, Germany; Natalie.kopplin@hotmail.de

**Keywords:** socioeconomic position, psychological capital, health literacy, health behaviors, mediation analysis, educational attainment, childhood socioeconomic position

## Abstract

Socioeconomic circumstances during childhood and adulthood are known to negatively affect health promoting behaviors. On the other hand, psychological capital (PsyCap) and health literacy are positively associated with these lifestyle behaviors. We, therefore, reasoned that PsyCap and health literacy might “buffer” the negative influences of socioeconomic circumstances on health-promoting behaviors. Method: We measured subjective childhood socioeconomic position (SEP) and adult educational attainment (as a proxy for adult socioeconomic circumstances), health literacy, PsyCap, and health behaviors (fruits and vegetables consumption, exercise, and sweets and cookies consumption) in a sample of N = 150 individuals (mean age 34.98 years, 66.7% female). Results: Bootstrapped mediation analyses including PsyCap and health literacy as parallel mediators revealed that: (I) The relationship between childhood SEP and (a) fruits and vegetables consumption and (b) exercise was mediated by PsyCap, and (II) the relationship between adult educational attainment and (a) fruits and vegetables consumption and (b) exercise was mediated by PsyCap and health literacy. We found no significant effects for consumption of sweets and cookies. Conclusion: These results suggest that larger studies are warranted that confirm the potential of PsyCap and health literacy in mitigating the negative effects of lower SEP on health behaviors and health outcomes.

## 1. Introduction

Childhood socioeconomic circumstances are known to affect a wide range of adult health outcomes, including mortality. Poulton et al. [1] established that compared to adults from higher childhood socioeconomic position (SEP) families, those with a lower childhood SEP reported poorer cardiovascular and dental health, and an increased likelihood of substance abuse. Similar findings were reported for the association between lower childhood SEP and adult inflammatory markers [2], smoking and mortality [3], increased risk for cancer [4], and overweight or obesity [5]. Importantly, many of these associations are not—or only slightly—mitigated by adjusting for upward mobility or adult SEP—e.g., [1,5]. Indeed, Galobardes et al. [6] established that despite adult social status, lower childhood SEP affects overall adult mortality, even in younger birth cohorts and in countries known as “welfare states” (i.e., Norway and Sweden). 

Evidence on the associations between childhood SEP and adult health behaviors is also accumulating, generally indicating that lower childhood SEP is associated with a range of harmful habitual behaviors in adulthood [7,8,9]. For example, Puolakka et al. [10] showed that lower childhood SEP is associated with several adult lifestyle factors, particularly with poorer diet, less physical activity, and increased risk of being a smoker, and Yanagi et al. [11] showed that lower childhood SEP is associated with decreased fruit and vegetable intake in older age. In addition to these cross-sectional studies, Hill et al. [12] provided experimental evidence that compared to those with higher childhood SEP, participants with lower childhood SEP consumed more snacks, even in the absence of hunger.

Efforts to explain the association between (childhood) SEP and adult health have focused on structural factors such as educational attainment [13], as well as on psychological or psycho-social variables as (partial) mediators, such as social support [14]. The reserve capacity model (RCM) [15] highlights the role of various “reserve capacities”—and in turn, positive and negative cognitions/emotions—as possible explanatory variables for the associations between SEP and health. Specifically, the model posits that the negative association between lower (childhood) SEP and health could be explained by lower levels of psychological reserves that allow individuals to cope with the stress and daily hassles of growing up and living in lower SEP environments, such as self-esteem, social support, and perceived control [15,16]. Interestingly, these protective effects seem to be especially present in lower SEP groups [17,18].

The aim of the current research is to investigate two additional variables that could protect individuals with a lower SEP against adverse health outcomes and health behaviors. Specifically, in the current research, we propose that one crucial protective variable that could explain the relationship between childhood SEP and adult health behaviors is psychological capital (PsyCap) [19]. PsyCap is a higher-order construct that consists of hope, efficacy, resilience, and optimism. Luthans et al. [19] (p. 3) define PsyCap as an “(…) individual’s positive psychological state of development, characterized by: (a) having confidence and skills to take on and put in the necessary effort to succeed at challenging tasks (efficacy); (b) making positive attributions about succeeding now and in the future (optimism); (c) persevering toward goals and, when necessary, redirecting paths to goals in order to succeed (hope); and (d) when beset by problems and adversity, sustaining and bouncing back and even beyond to attain success (resiliency)”. PsyCap is a consistent predictor of positive outcomes such as job performance, subjective well-being, and goal achievement in a variety of settings, including organizations and academia. PsyCap is increasingly being investigated and applied in (mental) health settings, and there are indications that both the individual components (hope, efficacy, resilience, and optimism) as well as the higher-order construct are associated with improved (mental) health and well-being. For example, Non et al. [20] report that optimism is associated with having a healthy diet and a healthy BMI, and Perna et al. [21] show that resilient individuals consumed more fruits and vegetables and were more likely to engage in exercise. Furthermore, higher levels of PsyCap are associated with lower BMI [22], and in our own research [16], we established that the relationship between adult educational attainment and self-reported health was (partly) mediated by PsyCap. Specifically, we found that individuals with a lower education also reported lower levels of PsyCap, and these, in turn, were associated with more health complaints and lower overall self-reported health.

Of course, healthy behaviors are more likely if one has knowledge of the benefits of, for instance, fruit and vegetable consumption or is aware of the risks of smoking and alcohol use, and if one can use this knowledge to make health-related decisions—i.e., healthy living is more likely if health literacy is high. The associations between SEP and health literacy have been established repeatedly. For example, in a recent study [23], high parental health literacy levels were predictive of healthier nutrition, increased physical activity, and regular tooth brushing in their children—and the only significant predictor of high health literacy was high SEP. Lower health literacy has, furthermore, consistently been associated with increased adult mortality, even after controlling for socioeconomic status [24]. Health literacy has also been shown to mediate between SEP—specifically, educational attainment –and self-reported health [25] and health behaviors [26]. Therefore, we will include health literacy as a second mediator in our analyses. 

The current research adds to—and differs from—existing literature on socioeconomic health inequalities by focusing on protective variables in the relationship between adverse socioeconomic circumstances and health behaviors. Moreover, our study will add to the literature on reserve capacities [15] by proposing that PsyCap is a psychological reserve that will enable individuals to cope with or overcome the adverse consequences of lower SEP. Thus, the current research will investigate whether PsyCap and health literacy mediate the relationship (a) between childhood SEP and health behaviors and (b) between adult educational attainment (as a proxy for adult SEP) and health behaviors. The results of this study may shed light on targets for health promotion efforts, especially given that PsyCap is malleable: immediate and longer-term (six months) increases in PsyCap levels have been reported after brief (1–3 h) workshops [27,28]. This makes PsyCap an interesting candidate to focus on when designing health promotion programs that focus on curbing the negative effects of lower childhood SEP on adult health outcomes. Similarly, increased health literacy makes optimal health decisions and behaviors more likely, and health literacy is another variable that is open to change [29]. Both variables will be considered as parallel mediators between socioeconomic circumstances and health behaviors.

## 2. Materials and Methods

### 2.1. Sample

We utilized a cross-sectional design. Data collection took place in April–May 2020. Power analysis indicated that to obtain power of 0.95, and assuming a (low) medium effect (f^2^ 0.10 to 0.15) with a significance level of α = 0.05, the required sample size should be n = 107~ 155 [30]. Since we wanted to obtain a sample that did not only include students, we chose to use snowball convenience sampling to obtain our sample, by spreading a social media post (via Facebook, LinkedIn, WhatsApp) that contained a link to the Qualtrics survey. In total, n = 202 individuals responded to the survey invitation. After exclusion of incomplete responses, the final sample consisted of n = 150 individuals (66.7% female, 76% German). The age of the participants ranged from 17 to 68 years, M = 34.98 (SD = 13.58). Of the participants, 46.7% had obtained a secondary school/A-level diploma, 31.3% had a bachelor’s degree (from college/university), 18.7% had a master’s degree, and 3.3% reported a PhD degree. The sample contained 30.7% students, 64.7% (self-) employed, and 2.7% unemployed individuals. The remaining 2% did not indicate an employment status.

### 2.2. Procedure and Measures

Participants received information about the study and provided informed consent after clicking on the link in the social media post. They, then, provided some demographics (age, gender, nationality, education, employment status, weight, and height (for BMI: kg/m^2^)), and continued with the following measures:

Relative childhood socioeconomic position. We used an established measure of relative childhood socio-economic position [12]. Participants were instructed to think back to their childhood before age 12 and answer the following questions on a 5-point scale (1 = strongly disagree, 5 = strongly agree): “My family had enough money for things growing up,” “I grew up in a relatively wealthy neighborhood,” and “I felt relatively wealthy compared to others my age”. Responses to these questions were aggregated (Cronbach’s α = 0.80).

Frequency of health behaviors. We asked participants to indicate on how many days of the week they (a) consumed five portions of fruit and vegetables and (b) exercised for at least 30 min, and (c) consumed sweets or cookies. Answers were given on a scale ranging from 1 (never) to 8 (7 days a week). 

Health literacy. Participants completed the 16-item European Health Literacy Survey Questionnaire (HLS-EU-Q16) [31]). Using a 7-point scale (1 = very easy, 7 = very difficult, participants responded to questions like: “How easy would you say it is to understand what the doctor says to you?” and “How easy would you say it is to understand health warnings about behavior such as smoking, low physical activity and drinking too much?”. Cronbach’s α = 0.91.

Psychological capital. As the last measure, participants completed the 12-item Compound PsyCap Scale (CPC-12) [32]. On a scale ranging from 1 (strongly disagree) to 7 (strongly agree) participants indicated their agreement with items such as “Right now, I see myself as being pretty successful” and “When I’m in a difficult situation, I can usually find my way out of it”. Cronbach’s α = 0.89.

After completing all measures, participants were debriefed about the study’s research questions, and thanked for their participation. The Ethics Research Committee of Psychology and Neuroscience at Maastricht University (ref. number 188_10_02_2018_S50) approved all materials.

## 3. Results

### 3.1. Correlations

First, a correlation table was created to investigate the necessity of controlling for age, BMI, or gender (in addition to education) in our main analyses, as well as the relationships between the core variables. This analysis revealed a significant association between childhood SEP and PsyCap (*r* = 0.20, *p* = 0.017), and significant associations between PsyCap, frequency of eating fruits and vegetables (*r* = 0.21, *p* = 0.011), frequency of exercise (*r* = 0.31, *p* < 0.01), and health literacy (*r* = 0.52, *p* < 0.001). Given that some of our main variables were associated with age, gender, education, and BMI, we decided to include these as covariates in further analyses. For a full overview of correlations, means, and SDs, see Table 1. 

### 3.2. Mediation Analyses

Following Hayes [33] and Zhao et al. [34], we continued with mediation analyses, even though the association between our independent and dependent variables was not present. (Note: Hayes [33] and other researchers argue that the total effect (path c, from X ➔ Y) should not be used as a “gatekeeper” for tests of mediation. Zhao et al. [34] refer to results such as reported here as indirect-only mediation: the mediated effect exists in the absence of a direct effect). We used the PROCESS macro (v3.5) [35], model 6 (5000 bootstraps) to test for indirect effects of two parallel mediators. In all analyses, our independent variables were childhood SEP or adult educational attainment, our mediators were PsyCap and health literacy, and our covariates were age, gender, and BMI. We conducted these analyses for frequency of eating five portions of fruits and vegetables, frequency of exercising for 30 min, and frequency of consumption of sweets and cookies. 

See Figure 1 for a visual overview, and see Table 2 for an overview of all results.

#### 3.2.1. Childhood SEP

##### Fruits and Vegetables

The full model was significant (F (6;141) = 3.53, *p* = 0.003), explaining 13% of the variance in frequency of consuming five portions of fruits and vegetables. Unique predictors were PsyCap (β = 0.94, t(141) = 3.27, *p* = 0.003) and age (β = −0.03, t(141) = −2.09, *p* = 0.04). We found evidence for indirect-only mediation for PsyCap: Childhood SEP ➔ PsyCap ➔ Fruits and vegetables, β = 0.14, SE = 0.07, 95% CI [0.014, 0.307]. There were no significant indirect effects for health literacy (β = −0.04, 95% CI [−0.162, 0.030]) or both mediators together (i.e., no parallel mediation; β = 0.05, 95% CI [−0.013, 0.135]). 

##### Exercise

The full model was significant (F (6;141) = 4.87, *p* < 0.001), explaining 17% of the variance in frequency of exercising 30 min per day. Unique predictors were PsyCap (β = 0.66, t(141) = 2.75, *p* = 0.007) and BMI (β = −0.10, t(141) = −2.46, *p* = 0.02). We found evidence for indirect-only mediation of PsyCap: Childhood SEP ➔ PsyCap ➔ Exercise, β = 0.09, SE = 0.05, 95% CI [0.006, 0.216]. There were no significant indirect effects for health literacy (β = 0.04, 95% CI [−0.022, 0.152]) or both mediators together (i.e., no parallel mediation; β = 0.04, 95% CI [−0.010, 0.111]). 

##### Sweets and Cookies

The full model was significant (F (6;141) = 2.26, *p* = 0.04), explaining 9% of the variance frequency of consumption of sweets and cookies. The only unique predictor was gender (β = 0.93, t(141) = 2.50, *p* = 0.01). We found no evidence for mediation of PsyCap (β = −0.03, SE = 0.04, 95% CI [−0.113, 0.051]), health literacy (β = −0.04, SE = 0.05, 95% CI [−0.158, 0.027]), or both mediators together (i.e., no parallel mediation; β = −0.01, SE = 0.02, 95% CI [−0.056, 0.018]). 

#### 3.2.2. Adult Educational Attainment

##### Fruits and Vegetables

The full model was significant (F (6;141) = 3.65, *p* = 0.002), explaining 13% of the variance in frequency of consuming five portions of fruits and vegetables. Unique predictors were PsyCap (β = 0.93, t(141) = 3.31, *p* = 0.001) and age (β = −0.03, t(141) = −2.16, *p* = 0.03). We found evidence for indirect-only mediation for PsyCap and health literacy together, i.e., for parallel mediation: Education ➔ Health literacy ➔ PsyCap ➔ Exercise, β = 0.08, SE = 0.04, 95% CI [−0.176, 0.046]. There were no significant indirect effects for only health literacy (β = −0.06, 95% CI [−0.162, 0.030]) or only PsyCap (β = 0.04, 95% CI [−0.056, 0.177]). 

##### Exercise

The full model was significant (F (6;141) = 4.96, *p* <.001), explaining 17% of the variance in frequency of exercising 30 min per day. Unique predictors were PsyCap (β = 0.63, t(141) = 2.67, *p* = 0.008) and BMI (β = −0.10, t(141) = −2.27, *p* = 0.02). We found evidence for indirect-only mediation of PsyCap and health literacy together, i.e., for parallel mediation: Education ➔ Health literacy➔ PsyCap ➔ Exercise, β = 0.06, SE = 0.03, 95% CI [.006, 0.135]. There were no significant indirect effects for only health literacy (β = 0.06, 95% CI [−0.028, 0.175]) or PsyCap (β = 0.03, 95% CI [−0.044, 0.122]).

#### Sweets and Cookies Consumption

The full model was significant (F (6;141) = 2.44, *p* = 0.03), explaining 9% of the variance frequency of consumption of sweets and cookies. The only unique predictor was gender (β = 0.98, t(141) = 2.62, *p* = 0.01). We found no evidence for mediation of PsyCap (β = −0.009, SE = 0.02, 95% CI [−0.061, 0.025]), health literacy (β = −0.07, SE = 0.06, 95% CI [−0.224, 0.028]), or both mediators together (i.e., no parallel mediation; β = −0.02, SE = 0.02, 95% CI [−0.069, 0.026]). 

## 4. Discussion

In the current research, we aimed to investigate the mediating role of PsyCap and health literacy on the relationships between (a) childhood socioeconomic circumstances and (b) adult educational attainment—as a proxy for adult SEP—and health behaviors. Childhood SEP provides a blueprint for the types of environment one is likely to encounter during adulthood and plays an important role in calibrating our physical and psychological responses to a range of (health-related) opportunities and challenges, including educational attainment. We reasoned that increased levels of psychological “reserves” [15] such as PsyCap and health literacy could mitigate the adverse effects of lower childhood SEP and adult education on health behaviors. We chose to focus on PsyCap given its focus on future-oriented emotions (hope, optimism) and goal-oriented cognitions (efficacy, resilience), and based on findings that higher levels of PsyCap empower individuals to take (back) control over their lives, including their health [19]. The predictive value of health literacy on health promoting behaviors and its association to childhood and adult socioeconomic circumstances has also been established [25,26]. Last but not least, both health literacy and PsyCap are open to development, making them ideal targets in health promotion efforts.

Our results provide evidence of indirect-only mediation. Using childhood SEP as an independent variable, our results showed that its relationship with fruits/vegetable consumption and exercise was fully explained by PsyCap. Health literacy did not act as a mediator, nor did we find parallel mediation, and we found no significant results for consumption of sweets and cookies. The correlations further showed that as expected, childhood SEP was positively associated with PsyCap, and PsyCap, in turn, was associated with most of the outcome variables. Health literacy was also associated with health behaviors, but not with childhood SEP. These findings are in line with previous research that established the protective role of PsyCap in the relationship between lower SEP and health outcomes [16] and between stress and physical and mental health problems [36].

Further, our analyses indicate that the relationship between adult educational attainment—as a proxy for adult SEP—and fruits/vegetables consumption and exercise was fully mediated by the combination of PsyCap and health literacy (parallel mediation), but we did not find evidence for single mediation. Again, we found no significant results for the consumption of sweets and cookies. The correlations indicated that educational attainment was associated with both PsyCap and health literacy, as well as with exercise. These findings fit in with research [37] that indicates that combining PsyCap and health literacy in an 8-week intervention increased health behaviors, compared to a control group. Interestingly, our findings suggest that mediation by PsyCap and health literacy was only present for beneficial behaviors, and not for undesirable behavior. However, in our questions about consumption of sweets and cookies, we did not specify the amount consumed per day (e.g., a whole roll of cookies versus only a single cookie), and this differentiation could have caused different results. Future research should carefully specify undesirable behaviors and, further, investigate the relationships between socioeconomic circumstances, PsyCap and health literacy, and other undesirable, or even harmful, behaviors such as smoking, alcohol use, or recreational drug use.

We found a strong positive association between PsyCap and health literacy, but the latter variable only mediated between adult educational attainment and health behaviors, and not for childhood SEP. This suggests that although they are related concepts, PsyCap and health literacy play different roles in individuals’ beneficial and risky health behaviors. Moreover, this finding also suggests that a solitary focus on increasing health literacy in lower SEP groups might not be sufficient to reduce health inequalities [38]. Knowledge about health and how to access healthcare are essential to increase health outcomes [39], but that additional efforts need to be made to provide individuals with the ability and the belief that they are able to attain certain health-related goals. PsyCap may be able to buffer for the negative association between a lower childhood SEP and adult health behaviors, since this association is explained by a low level of PsyCap in adulthood. We, therefore, argue that this is where PsyCap can play a pivotal role, since increased levels of PsyCap create a future-oriented cognitive and emotional outlook [40] and are related to problem-oriented coping styles [41]. Together, these two variables might provide individuals with the tools needed to take control over their own health and health behaviors.

The current study has some limitations that need to be acknowledged. First, we present cross-sectional data, and although childhood SEP was assessed in a retrospective way, and our hypothesized patterns did emerge, it is not possible to draw causal conclusions. Further, the snowball sampling resulted in a convenience sample of participants that were generally quite highly educated and reported a middle-high to high childhood SEP. We, therefore, caution the generalization of our findings to the larger population, also in light of our relatively small sample size. Future research should take efforts to include a more stratified sample, specifically since increasing variance in the predictor variables increases the power to find significant effects. Moreover, it may be worthwhile to take the accessibility and availability of fresh produce (and fast food outlets) into account when studying healthy eating patterns, since individuals are likely to consume more fruits and vegetables if these are accessible and available [36]. Further, we utilized a measure of subjective relative childhood SEP, assessed retrospectively, and we have no access to more objective assessments of our participants’ childhood and adult SEP. Although some researchers have criticized retrospective assessments of childhood SEP, others—e.g., [42] (p. 32)—posit that they “(…) provide a useful opportunity to examine theoretical life course models empirically in the absence of complete data across the life course”. Furthermore, subjective social status—i.e., one’s perceived standing in the social hierarchy compared to one’s peers—is increasingly considered in research on health inequalities and has been associated with a range of health outcomes [43]. Lastly, it is known that (chronic) stress levels influence the relationship between socioeconomic circumstances and health outcomes (including health behaviors), most likely by depleting one’s psychological reserves [16]. However, we did not include a stress measure in the current research, so we cannot determine whether stress would moderate the mediation patterns we describe here.

## 5. Conclusions

The current research examined the mediating role of PsyCap in the relationship between childhood SEP and adult health behaviors and between adult educational attainment and health behaviors. Our findings suggest that increasing individuals’ PsyCap and health literacy may positively affect health behaviors, and as such, these variables could buffer against adverse childhood and adult socioeconomic circumstances. These findings are in line with the reserve capacity model [15], which predicts that such reserves can help mitigate the adverse effects of lower socioeconomic circumstances on health outcomes. Further, both PsyCap and health literacy are open to development, making them ideal targets for health promotion efforts. Our findings also indicate that such health promotion efforts should ideally be undertaken during a time when (un)healthy habits are forming, i.e., during childhood and adolescence [44].

## Figures and Tables

**Figure 1 ijerph-18-09399-f001:**
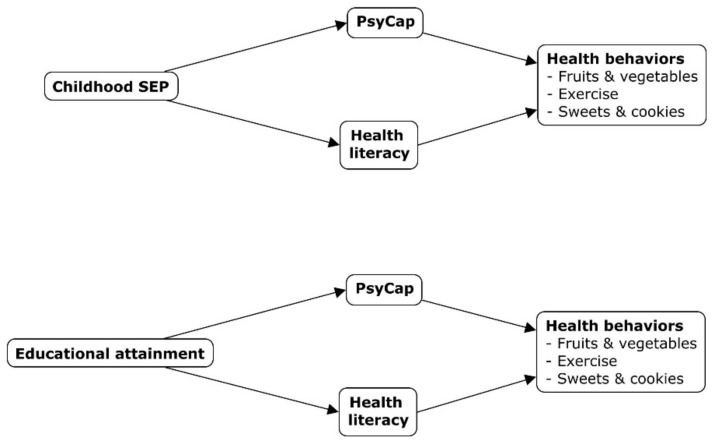
Schematic overview of conducted mediation analyses.

**Table 1 ijerph-18-09399-t001:** Descriptives and correlations of variables included in the current research (n = 150).

Variable	Mean (SD)	1	2	3	4	5	6	7	8	9
1. Childh. SEP	3.49 (0.94)	--								
2. Health lit.	5.30 (0.80)	0.12								
3. PsyCap	5.43 (0.80)	0.20 *	0.52 **							
4. Fruit/veg.	4.36 (2.38)	0.11	0.04	0.21 *						
5. Exercise	4.23 (2.06)	0.06	0.27 **	0.32 **	0.26 **					
6. Cookies	4.32 (2.07)	−0.04	−0.19 *	−0.15	−0.07	−0.20 *				
7. Age	34.98 (13.58)	−0.19 *	0.08	0.18 *	−0.18 *	0.07	0.07			
8. Gender	--	0.04	−0.10	−0.17 *	0.15	−0.09	0.22 **	−0.16		
9. BMI	24.25 (4.19)	−0.07	0.004	0.01	−0.18 *	−0.19 *	0.004	0.32 **	−0.29 **	
10. Adult Educ.	--	0.04	0.25 **	0.20 *	0.11	0.18 *	0.002	−0.02	−0.11	−0.16

*Note*. * *p* < 0.05, ** *p* < 0.01; gender 1 = male, 2 = female.

**Table 2 ijerph-18-09399-t002:** Summary of mediation analyses; using childhood SEP and adult education as independent variables and PsyCap and health literacy as (parallel) mediator variables.

	DirectEffect ^a^	Adjusted Direct Effect ^b^	R^2^	Mediator(s)	Indirect Effect ^c^	Indirect Effect 95% CI ^d^
Childhood SEP						
Fruits and vegetables	0.18	0.03	0.13	PsyCap	0.14	0.014 to 0.307
				Health literacy	−0.04	−0.162 to 0.029
				Health literacy and PsyCap	0.05	−0.013 to 0.135
Exercise	0.11	−0.07	0.17	PsyCap	0.10	0.006 to 0.216
				Health literacy	0.04	−0.022 to 0.152
				Health literacy and PsyCap	0.04	−0.010 to 0.111
Sweets and cookies	−0.08	−0.003	0.09	PsyCap	−0.03	−0.113 to 0.051
				Health literacy	−0.04	−0.158 to 0.027
				Health literacy and PsyCap	−0.01	−0.056 to 0.018
Adult Education						
Fruits and vegetables	0.21	0.15	0.13	PsyCap	0.04	−0.056 to 0.177
				Health literacy	−0.06	−0.176 to 0.046
				Health literacy and PsyCap	0.08	0.017 to 0.176
Exercise	0.27	0.13	0.17	PsyCap	0.03	−0.044 to 0.122
				Health literacy	0.06	−0.028 to 0.175
				Health literacy and PsyCap	0.06	0.006 to 0.135
Sweets and cookies	0.07	0.17	0.09	PsyCap	−0.009	−0.061 to 0.025
				Health literacy	−0.07	−0.224 to 0.028
				Health literacy and PsyCap	−0.02	−0.069 to 0.026

Note. Covariates in the model: Age, gender, BMI. ^a^ Effect of childhood SEP/adult education on the dependent variable. ^b^ Effect of childhood SEP/adult education on the dependent variable, adjusted for influence of the mediator Psychological Capital (PsyCap). ^c^ Indirect effect of childhood SEP/adult education on the dependent variable via the mediator variables PsyCap and Health literacy. ^d^ Based on 5000 bootstrap samples of n = 150; indicates a significant effect if the CI does not contain zero.

## Data Availability

Syntax, data, and survey instrument are available on: https://osf.io/pfg6n/ (accessed on 2 September 2021).

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
