# Peer review of "Childhood Socioeconomic Position, Adult Educational Attainment and Health Behaviors: The Role of Psychological Capital and Health Literacy"

_ijerph, 2021, doi:10.3390/ijerph18179399_

Round 1
Reviewer 1 Report
#Comments to authors
The manuscript, entitled “Childhood socioeconomic position, adult educational attainment and health behaviors: The role of psychological capital and health literacy.” This is an interesting study corresponding to the problems of public health of many countries, particularly children and adults in developing countries whose SEP is relatively low.
- Introduction
- The authors provided clear information about what has been done; however, it would make more sense when the authors could indicate how this study’s findings would make a significant difference from existing studies.
- Methods:
- It was not clear how the authors selected the respondents and why that number.
- The authors should classify the samples by spatial distribution (residential location) because those who live in an area where is far from the market/farm may have less behavioral intention to buy and eat fruits/vegetables. Accessibility and availability of fruits/vegetables in the neighborhoods significantly influence the daily consumption.
- Discussion
- Discussion is strongly enough; however, it would be more interesting to see how authors use their findings to argue/with existing studies and then inform readers that how those findings are significantly different and important.
- Conclusion
- L307: the authors claimed that this research was “among the first to examine the mediating role of PsyCap…”, but we did not see any evidence in the manuscript that supported your claim. The authors should explain it somewhere in the literature(introduction) or discussion section before coming up to this conclusion.
Author Response
Thank you for your constructive comments on our manuscript. Please see the attached document for a point-by-point response.

Reviewer 2 Report
Dear authors,
Your manuscript is interesting but I need you to answer some questions:
INTRODUCTION
- The authors have not stated the objective of the investigation.
MATERIALS AND METHODS
Design:
- The authors must specify the research design.
Sample:
- What was the target population? The authors must specify it.
- The authors must include the response rate of the participants in the study.
REFERENCES
- Many bibliographies are obsolete. The bibliographic citations used are more than 5 years old (48.8 %). The authors must update and arrange the bibliography.
- The authors have mixed APA and Vancouver citation regulations. You must write the references correctly.
Author Response
Thank you for your constructive comments. Please see the attached document for a point-by-point response.

Round 2
Reviewer 2 Report
Dear authors,
Thanks for your reply. The explanations that you provide are satisfactory. The paper has greatly improved its quality.
Congratulations on your work.
Best regards
Author Response
Thank you very much for your compliments!
kind regards, the authors